# CRISPR-Cas9 Approach Constructed Engineered *Saccharomyces cerevisiae* with the Deletion of *GPD2*, *FPS1*, and *ADH2* to Enhance the Production of Ethanol

**DOI:** 10.3390/jof8070703

**Published:** 2022-07-01

**Authors:** Peizhou Yang, Shuying Jiang, Suwei Jiang, Shuhua Lu, Zhi Zheng, Jianchao Chen, Wenjing Wu, Shaotong Jiang

**Affiliations:** 1Anhui Key Laboratory of Intensive Processing of Agricultural Products, College of Food and Biological Engineering, Hefei University of Technology, Hefei 230009, China; jiangshuying9805@163.com (S.J.); lsh2641715623@163.com (S.L.); zhengzhi@hfut.edu.cn (Z.Z.); cjc183030@163.com (J.C.); wwj8121@163.com (W.W.); jiangshaotong@163.com (S.J.); 2Department of Biological, Food and Environment Engineering, Hefei University, Hefei 230601, China; jiangsw@hfuu.edu.cn

**Keywords:** *Saccharomyces cerevisiae*, CRISPR-Cas9, ethanol, gene knockout, by-product formation, transcriptome analysis

## Abstract

Bioethanol plays an important value in renewable liquid fuel. The excessive accumulation of glycerol and organic acids caused the decrease of ethanol content in the process of industrial ethanol production. In this study, the CRISPR-Cas9 approach was used to construct *S. cerevisiae* engineering strains by the deletion of *GPD2*, *FPS1*, and *ADH2* for the improvement of ethanol production. RNA sequencing and transcriptome analysis were used to investigate the effect of gene deletion on gene expression. The results indicated that engineered *S. cerevisiae* SCGFA by the simultaneous deletion of *GPD2*, *FPS1*, and *ADH2* produced 23.1 g/L ethanol, which increased by 0.18% in comparison with the wild-type strain with 50 g/L of glucose as substrate. SCGFA strain exhibited the ethanol conversion rate of 0.462 g per g of glucose. In addition, the contents of glycerol, lactic acid, acetic acid, and succinic acid in SCGFA decreased by 22.7, 12.7, 8.1, 19.9, and 20.7% compared with the wild-type strain, respectively. The up-regulated gene enrichment showed glycolysis, fatty acid, and carbon metabolism could affect the ethanol production of SCGFA according to the Kyoto Encyclopedia of Genes and Genomes (KEGG) analysis. Therefore, the engineering strain SCGFA had great potential in the production of bioethanol.

## 1. Introduction

Ethanol, an alternative energy to fossil fuels, relieves the double pressures of energy crisis and environmental protection [1]. As an ethanol-producing strain, *Saccharomyces cerevisiae* consumes glucose to produce bioethanol through various biochemical reactions with the following steps: (1) glucose is decomposed into pyruvate through the glycolysis pathway; (2) the pyruvate is catalyzed by a decarboxylase to produce carbon dioxide and acetaldehyde; and (3) the acetaldehyde is further catalyzed into ethanol by alcohol dehydrogenase [2]. However, ethanol production is generally accompanied by the formation of by-product during the fermentation of *S. cerevisiae* [3]. The excessive by-product formation definitely caused the efficiency decrease of glucose conversion [4]. Therefore, the by-product formation in *S. cerevisiae* reduced product quality and increased operating procedures, thus increasing the production costs.

Glycerol, a main by-product in *S. cerevisiae*, maintained important physiological functions in osmotic pressure balance and intracellular redox balance [5,6,7]. However, the excessive accumulation of glycerol undoubtedly reduced the conversion rate of glucose. During the glycerol metabolism, glycerol 3-phosphate dehydrogenase 2 (Gpd2) was responsible for the glycerol synthesis [8]. In addition, GPD2 also played a crucial role in the redox balance under anaerobiosis [9]. The deletion of *S. cerevisiae GPD2* caused the decrease of the synthesis efficiency of glycerol [5]. Thus, *GPD2* gene modification was an effective way to redirect the carbon flux of glycerol synthesis.

Glycerol uptake experiments indicated glycerol generally consisted of an FPS1-independent component, which facilitated diffusion based on the permeability characteristics of yeast plasma membrane [10]. Aquaglyceroporin FPS1p, a member of the major intrinsic protein (MIP) family of channel proteins, was a facilitator for glycerol uptake and efflux in response to the extracellular changes in *S. cerevisiae* [10,11]. The overexpression of *FPS1* enhanced the glycerol production [10]. In contrast, *FPS1* depletion prevented the constitutive glycerol release by blockage of the secretory pathway [12]. In addition, the industrial ethanol production was controlled with high-level ethanol accumulation under anaerobic conditions [13]. Under these conditions, ethanol was generally catalyzed into acetaldehyde by alcohol dehydrogenase 2 (Adh2) [14]. The *ADH2* deletion increased the ethanol titer and yield during the fermentation processing of *S. cerevisiae* [15]. Therefore, *GPD2*, *FPS1*, and *ADH2* deletion could improve the ethanol yield of *S. cerevisiae* by redirection of the metabolic pathways.

However, the single-gene deletion was difficult to improve the overall yield of ethanol because *GPD2*, *FPS1*, and *ADH2* played different roles in the metabolic mechanism of *S. cerevisiae*. Recently, although multiplex genome engineering has been developed to disrupt the target genes in *S. cerevisiae* [16,17,18,19], single genetic locus deletion is still an effective way to knock out the target gene [20]. The effect of the combinations of *GPD2*, *FPS1*, and *ADH2* deletion on the ethanol yield in *S. cerevisiae* has still not been investigated so far. In this study, four combinations of *GPD2*, *FPS1,* and *ADH2* deletion in *S. cerevisiae* were investigated using the clustered regularly interspaced short palindromic repeats-Cas9 (CRISPR-Cas9) approach (Figure 1). The ethanol yields of four *S. cerevisiae* engineering strains ware compared. In addition, the contents of glycerol, organic acids, and carbon dioxide (CO_2_) were also determined to investigate the effect of gene deletion on the formation of by-products. This study constructed *S. cerevisiae* engineering strains for improvement of bioethanol yield based on the comprehensive considerations of by-product formation and ethanol consumption.

## 2. Materials and Methods

### 2.1. Plasmids, Primers, and Agents

Plasmids gRNA-trp-HyB and Cas9-NTC were from Addgene Company (Watertown, MA, USA). Plasmids gRNA-trp-HyB for guide RNA synthesis and Cas9-NTC for *S. cerevisiae* genome DNA digestion carried hygromycin B and nourseothricin resistance genes, respectively. Primers and genes were synthesized by Sangon Biotech (China). The gene sequencing was also performed by Sangon Biotech (Shanghai, China). Enzymes such as Phusion high-fidelity PCR master mix and *Thermus aquaticus* DNA polymerase were from NEB Biotech (Ipswich, MA, USA). Other reagents such as nourseothricin, hygromycin B, polyethylene glycol (PEG), salmon sperm DNA (ssDNA), glucose, yeast extract, and peptone were from Trans Gen Biotech (Beijing, China). All chemical reagents are analytically pure.

### 2.2. Linear Vector Construction for gRNA Synthesis

The gRNA sequences of 20 bp for *S. cerevisiae GPD2*, *FPS1*, and *ADH2* were searched using the Weblink http://chopchop.cbu.uib.no/system (accessed on 2 May 2020) online search [21]. The target sequences were selected according to the efficiency and self-complementarity values. Three different gRNA expression vectors were obtained by the amplification of plasmid gRNA-trp-HyB using the designed primers (Table 1). The different pairs of primers were used to amplify three different gRNA vectors by Phusion High-Fidelity PCR Master Mix as the following parameters of 2× Phusion master mix 12.5 μL, forward and reverse primers of 1.25 μL, DNA template of 0.5 μL, and nuclease-free water of 9.5 μL. The size of each gRNA vector was 6509 bp. The processing parameters of PCR amplification were 98 °C for 30 s; 98 °C for 8 s; 50 °C for 25 s; 72 °C for 3 min for 29 cycles; and 72 °C for 10 min.

### 2.3. S. cerevisiae GPD2, FPS1, and ADH2 Knockout by CRISPR-Cas9 Approach

The CRISPR-Cas9 approach was used to construct *S. cerevisiae* engineering strains according to the technical solutions shown in Figure 2. The transformation of exogenous genes into *S. cerevisiae* was performed according to the PEG-mediated LiA-ssDNA method [22]. The target gene was knocked out by the integration of insertion DNA into *S. cerevisiae* genome DNA by the CRISPR-Cas9 approach [23]. Firstly, Cas9-NTC plasmid was transformed into *S. cerevisiae* based on the resistance screening of nourseothricin. After transformation, the solution of 50 μL was incubated on the solid yeast extract peptone dextrose medium (YPD) plate containing 80 μg/mL of nourseothricin. After culture at 30 °C for 48 h, the putative colonies were screened out. The true transformants were named *S. cerevisiae*-Cas9-NTC after identification. Secondly, gRNA vector and insertion DNA were transformed into *S. cerevisiae*-Cas9-NTC by the PEG-mediated LiA-ssDNA method [22]. After transformation, the solution of 50 μL was cultured on the solid YPD plate containing 80 μg/mL of nourseothricin and 300 μg/mL of hygromycin B. After cultivation at 30 °C for 48 h, the putative colonies were screened out for further identification.

### 2.4. Identification of Engineering Strains by DNA Amplification and Sequencing

The constructed vectors of gRNA-*GPD2*, gRNA-*FPS1*, and gRNA-*ADH2* were used to recognize the target sites of *GPD2*, *FPS1*, and *ADH2*, respectively. *GPD2*, *FPS1*, and *ADH2* were knocked out by the insertion of 2091 bp of *TV-AFB1D*, 879 bp of *OM-PLA1*, and 910 bp of *DPE* in *S. cerevisiae*, respectively. Three different pairs of primers based on the sequences of *TV-AFB1D*, *OM-PLA1*, and *DPE* genes were designed to amplify insertion DNA for integration identification (Table 1). After sequencing confirmation, the true transformants were identified and used for further research.

### 2.5. Growth Determination of Engineered S. cerevisiae

The optical density (OD) at a wavelength of 600 nm was used to determine the cell concentrations of *S. cerevisiae* to investigate the effect of gene knockout on the cell growth of engineered *S. cerevisiae*. The amount of 1 mL of *S. cerevisiae* fermentation broth with 1 OD_600nm_ was sucked out and then inoculated into a 250-mL conical flask containing 100 mL YPD to culture at 30 °C with a shaking speed of 200 rpm. The OD_600nm_ values were measured every 6 h during the fermentation of 72 h.

### 2.6. Sample Treatment for the Content Measurement of By-Products

The glucose and ethanol concentrations were measured to investigate the relation of glucose consumption with ethanol production. The *S. cerevisiae* engineering strains were inoculated into a 250-mL conical flask containing 100 mL of YPD medium containing 50 g/L of glucose when the cell concentration reached 1 OD_600nm_. After fermentation for 24 h, the fermentation solution was transferred into an anaerobic fermentation condition. The fermentation solution was sucked out for the detection of ethanol and glucose contents every 6 h during the fermentation of 72 h. In addition, the concentrations of glycerol and organic acids were also investigated. The organic acids in the broth mainly included succinic acid, acetic acid, and lactic acid in this study. The CO_2_ concentration was determined using a CO_2_ online detector manufactured by Hengxin Company (Dongguan, China).

### 2.7. Determination of By-Products by the HPLC Method

The concentrations of glucose, ethanol, and glycerol were measured by high-performance liquid chromatography (HPLC) [24]. The operation parameters are a mobile phase of 0.01 mol/L H_2_SO_4_, column temperature of 50 °C, instruments of Waters 1525 Binary HPLC Pump, Waters 2410 Refroctive Index Detector, and Shodex SH1011 chromatographic column. The concentrations of organic acids were determined according to the HPLC method [25]. The parameters were a detection wavelength of 210 nm, mobile phase A of 10 mM KH_2_PO_4_, mobile phase B of methanol, flow rate of 1.0 mL/min, and column temperature of 30 °C, instruments of Waters Alliance E2695, Waters 2489 UV detector, and Waters XSelect HSS column.

### 2.8. Transcriptome Analysis

Transcriptome analysis of *S. cerevisiae GPD2 delta FPS1 delta ADH2 delta* was carried out based on the data of RNA sequencing using the wild-type strain as the control. Total RNA was extracted after a fermentation for 24 h with glucose as the carbon source [26]. The concentration of extracted RNA was measured using a Qubit^®^ RNA Assay Kit made by Life Technologies Corporation (Gaithersburg, MD, USA). RNA purity was detected using a Nano Photometer Spectrophotometer manufactured by IMPLEN Company (Munich, Germany). RNA integrity was assessed using an RNA Nano 6000Assay Kit made by Agilent Technologies Company (Santa, CA, USA). Sangon Biotech Company (Shanghai, China) constructed the RNA-Seq libraries and sequenced using the Illumina Hiseq 2500 platform [27]. *S. cerevisiae* S288C (assembly R64) was selected as the reference genome for the transcriptomic profiling.

### 2.9. Data Analysis

All statistics data were mean ± standard error by three repetitions. The curve figures were drawn using OriginPro2018 Software (version number GF3S4-9089-7991320).

## 3. Results

### 3.1. Engineered S. cerevisiae Construction

The 20-bp of gRNA sequences for *GPD2*, *FPS1*, and *ADH2* knockout were chosen on the online search platform of weblink http://chopchop.cbu.uib.no/ (accessed on 2 May 2020). The efficiencies of *GPD2*, *FPS1*, and *ADH2* deletion were 70.6, 70.4, and 70.9%, respectively (Table 1). *S. cerevisiae* engineering strains were transformed by Cas9-NTC and gRNA vectors using insertion DNA as an exogenous donor DNA. In this study, engineered *S. cerevisiae* strains with *GPD2Δ*, *FPS1Δ*, *GPD2Δ FPS1Δ*, and *GPD2Δ FPS1Δ ADH2Δ* were named by SCG, SCF, SCGF, and SCGFA, respectively. Four *S. cerevisiae* engineering strains of SCG, SCF, SCGF, and SCGFA were constructed according to the technology step (Figure 3A). The putative SCGFA colony was screened on the solid plates containing double antibiotics of nourseothricin and hygromycin B (Figure 3B). PCR amplification was used to identify SCGFA using the genome DNA as a template. The DNA bands with the sizes of 2091, 910, and 879 bp indicated *TV-AFB1D*, *DPE*, and *OM-PLA1* as insertion DNA, respectively (Figure 3C). The true SCGFA transformants were confirmed by gene sequencing. The other three transformants of SCG, SCF, and SCGF were also confirmed by DNA amplification and gene sequencing.

### 3.2. Effect of Gene Deletion on the Proliferation of S. cerevisiae

The OD_600nm_ values of the wild-type and four *S. cerevisiae* engineered strains were determined to investigate the effect of gene knockout on the cell growth of engineered strains (Figure 4). After fermentation for 72 h, the OD_600nm_ values of the wild-type strain, SCG, SCF, SCGF, and SCGFA were 9.83, 9.59, 9.47, 9.64, and 9.77, respectively. The growth rates in the logarithmic phase of yeast proliferation were 0.4825, 0.4463, 0.4503, 0.4510, and 0.4720, respectively. The OD_600nm_ value and logarithmic phase growth rates of four genetically engineered *S. cerevisiae* were slightly lower than those of the wild-type strain after fermentation for 72 h. Thus, the gene knockouts in *GPD2*, *FPS1*, and *ADH2* loci did not significantly affect the cell proliferation of *S. cerevisiae* engineered strains.

### 3.3. Effect of Gene Knockout on the Glucose Consumption

The residual contents of glucose were determined to investigate the efficiency of glucose consumption by engineered *S. cerevisiae* after gene knockout (Figure 5). The glucose was almost consumed after fermentation for 48 h using the initial glucose contents of 50 g/L. Four engineered strains of *S. cerevisiae* have a similar change trend to the wild-type strain under a batch fermentation. The gene deletion in three loci of *GPD2*, *FPS1*, and *ADH2* did not affect the consumption of glucose.

### 3.4. Effect of Gene Deletion on the Ethanol Production

The ethanol contents were measured to investigate the effect of gene deletion on the ethanol production of *S. cerevisiae* engineering strains. The ethanol contents of engineering strains had a similar change trend to the wild-type strain during the initial fermentation of 0–24 h. However, the ethanol contents of strains exhibited a remarkable difference during the subsequent fermentation of 24–48 h. The ethanol contents from engineering strains exceeded the wild-type strain. The ethanol contents of strains kept relatively stable during the final fermentation of 48–72 h. the highest ethanol contents of SCG (20.6 g/L), SCF (20.9 g/L), SCGF (22.2 g/L), and SCGFA (23.1 g/L) were 1.05, 1.07, 1.13, and 1.18-fold compared with the wild-type *S. cerevisiae* (19.6 g/L), respectively. The ethanol conversion rate of SCGFA was 0.462 g per g of glucose, which was higher than the wild-type strain (0.392 g ethanol per g of glucose). Thus, the SCGFA strain constructed by triple-deletion *GPD2*, *FPS1*, and *ADH2* obtained a higher yield of ethanol than the single or double-deletion approaches.

### 3.5. Glycerol Production of Engineered S. cerevisiae

The glycerol contents in the fermentation broth were determined to compare the difference among the *S. cerevisiae* engineering strains (Figure 6). All the glycerol contents of SCG, SCF, SCGF, and SCGFA were lower than the wild-type *S. cerevisiae* during the fermentation. After fermentation for 72 h, SCG, SCF, SCGF, and SCGFA obtained 1787, 1729, 1677, and 1738 mg/L of glycerol in broth, respectively, which decreased by 20.5, 23.1, 25.4, and 22.7% compared with the wild-type strain (2249 mg/L). The glycerol contents from four different engineering strains decreased due to the gene deletion under different combinations. The SCGF strain with *GPD2* and *FPS1* deletion represented the lowest glycerol content in broth among four engineering strains.

### 3.6. Lactic Acid Production of S. cerevisiae

The contents of lactic acid were measured to investigate the effect of gene knockout on lactic acid production in *S. cerevisiae* (Figure 7). The lactic acid contents of four engineered strains were lower than the wild-type *S. cerevisiae*. The lactic acid contents of SCG (7.22 mg/L), SCF (6.38 mg/L)*,* SCGF (6.88 mg/L), and SCGFA (6.59 mg/L) decreased by 4.4, 15.5, 8.9, and 12.7% compared with the wild-type strain (7.55 mg/L), respectively. Therefore, the deletion of *S. cerevisiae GPD2*, *FPS1*, and *ADH2* resulted in the decrease of lactate content.

### 3.7. Effect of Gene Knockout on the Production of Acetic Acid

The contents of acetic acid in the fermentation broth were measured to investigate the effect of gene deletion with different combinations on the acetic acid production during the fermentation (Figure 8). The acetic acid contents in four engineering strains were lower than the wild-type *S. cerevisiae*. After fermentation for 72 h, the acetic acid contents of SCG (116 mg/L), SCF (112 mg/L), SCGF (115 mg/L), and SCGFA (114 mg/L) in fermentation broth decreased by 6.5, 9.7, 7.3, and 8.1% compared with the wild-type *S. cerevisiae* (124 mg/L), respectively. The different combinations of *GPD2*, *FPS1*, and *ADH2* deletions led to the decrease of acetic acid content in engineered *S. cerevisiae* mutants.

### 3.8. Succinic Acid Production during the Fermentation of S. cerevisiae

The succinic acid concentrations were measured to investigate the gene deletion with different combinations on succinic acid production (Figure 9). The succinic acid concentrations from four *S. cerevisiae* engineering strains were lower than the wild-type *S. cerevisiae*. The succinic acid concentrations from SCG (17.3 mg/L), SCF (16.3 mg/L), SCGF (16.2 mg/L), and SCGFA (15.7 mg/L) decreased by 11.73, 16.84, 17.35, and 19.9% compared with the wild-type *S. cerevisiae* (19.6 mg/L), respectively. Thus, the concentration of succinic acid decreased in all the tested engineered *S. cerevisiae* mutants.

### 3.9. Comparison of CO_2_ Concentrations of the Wild-Type and Engineered Strains

The concentrations of CO_2_ released from different *S. cerevisiae* strains were investigated by the online detection approach (Figure 10). Four engineered strains exhibited lower CO_2_ concentrations compared with the wild-type *S. cerevisiae*. The CO_2_ concentrations from SCG (1011 mg/L), SCF (956 mg/L), SCGF (924 mg/L), and SCGFA (897mg/L) decreased by 10.6, 15.5, 18.3, and 20.7% compared with the wild-type *S. cerevisiae* (1131 mg/L). The *GPD2*, *FPS1*, and *ADH2* deletion resulted in the decrease of CO_2_ concentrations released from four engineering *S. cerevisiae* strains. Thus, the gene deletion had a dramatic effect on the respiratory metabolism of *S. cerevisiae* based on the amount of CO_2_ emission.

### 3.10. Stability of Ethanol Production by SCGFA Engineering Strain

The contents of ethanol by SCGFA engineering strains after multiple generations of culture were measured to analyze the stability of ethanol production (Figure 11). The contents of ethanol of SCGFA engineering strain after the 1st, 10th, 20th, 30th, 40th, and 50th generations were close to 23 g/L using 50 g/L glucose as fermentation substrate, which was higher than the wild-type strain (19.6 g/L). The results indicated that the SCGFA engineering strain could steadily produce ethanol after several generations. Therefore, the SCGFA engineering strain constructed by the CRISPR-Cas9 approach still maintained the stable capability of ethanol production after gene deletion.

### 3.11. Carbon Balance Analysis

After fermentation for 72 h, the contents of metabolites in the broth tended to be stable according to the data in Figure 4. The carbon balances of each strain were analyzed based on the contents of ethanol and the main by-products (Table 2). In this study, the concentration of 50 g/L glucose was converted into 1.66667 C·mol/L based on the molar mass of carbon. The results showed the carbon content of ethanol in SCGFA was higher than those in other strains. In addition, the total carbon content in SCGFA was 1.09128 C·mol/L, which was 1.15-fold in comparison with the wild-type strain. Thus, SCGFA exhibited a high ethanol conversion capacity.

### 3.12. Differentially Expressed Genes (DEGs)

RNA sequencing analysis was carried out to explore the effect of gene knockout on the gene expression of the *S. cerevisiae* engineering strain (Figure 12). A total of 570 DEGs were identified in the SCGFA engineered strain compared to the wild-type strain, in which 166 and 404 genes belonged to up-regulated and down-regulated genes according to the log_2_ (Transcripts Per Million, TPM) values, respectively. In addition, quite a few not differential expressed genes existed in the detected genes.

### 3.13. Function Annotation of Differential Genes

According to the gene ontology (GO) analysis, 570 DEGs were divided into three categories including biological processes, cellular components, and molecular functions (Figure 13). Both the up-regulated genes and down-regulated genes were enriched in all three categories. Overall, the proportion of down-regulated genes was much higher than that of up-regulated genes. In the cellular component, the proportion of up-regulated genes related to membrane accounted for 3.11% of the total number of genes. In the molecular function, the proportion of up-regulated genes related to the antioxidant activity approximately accounted for 2.8%. Thus, the enrichment analysis of up-regulated genes showed that the SCGFA strain could maintain the stability of cell growth, enhance the stability of cell membrane, and improve the osmotic balance during fermentation.

In the biological process, the down-regulated genes related to bio-adhesion and cell aggregation all accounted for more than 20% of the corresponding classification. In the cellular components, the down-regulated genes had no remarkable effect on the nucleolus, cell junction, and extracellular part. The down-regulated genes with supramolecular fibers, membrane-sealed lumens, and protein-containing complexes accounted for more than 10% of the total genes. For molecular functions, the down-regulated genes were mainly enriched in transcription factor activity, protein binding, electron transfer activity, translation regulation activity, and molecular transducer activity. Therefore, the results of down-regulated gene enrichment showed the simultaneous knockout of *GPD2*, *FPS1*, and *ADH2* might affect the transport of glycerol in the cell membrane of *S. cerevisiae*.

### 3.14. KEGG Analysis

The Kyoto Encyclopedia of Genes and Genomes (KEGG) database was used to analyze the DEGs in the SCGFA strain. The enrichment of up-regulated genes of the top 30 enrichment pathways are shown in Figure 14A. The main enrichment pathways of up-regulated genes were energy metabolism (such as glycolysis, gluconeogenesis, glyoxylic acid, dicarboxylic acid metabolism, fatty acid, and carbon), amino acid metabolism, and important life substance metabolism (such as purine metabolism, inositol phosphate metabolism, amino acid sugar, and nucleotide sugar metabolism, etc.). Thus, the up-regulated gene enrichment showed that glycolysis, fatty acid, and carbon metabolism could affect the ethanol formation of the SCGFA strain.

The top 30 enrichment pathways of down-regulated genes were shown in Figure 14B. The down-regulated genes were mainly related to mRNA monitoring pathway, eukaryotic ribosome biogenesis, RNA degradation, RNA transporters, non-homologous terminal junction, meiosis yeast, homologous recombination, cell cycle yeast, apoptosis multiple species, MAPK signaling pathway-yeast, and other pathways. The down regulation of these genes indicated that the simultaneous knockout of *GPD2*, *FPS1*, and *ADH2* could have a certain impact on the growth and proliferation of *S. cerevisiae*.

## 4. Discussions

Ethanol is likely to play an important role in the development of renewable fuel [28]. During the fermentation for ethanol production, glycerol was generally formatted as a byproduct to maintain osmotic stress and prevent water loss under hyperosmotic conditions [29]. However, the excessive accumulation certainly caused an adverse impact on ethanol production. The gene deletion has been proved to be an effective way to increase the ethanol yield with the reduction of glycerol production in the previous reports (Table 3). The redirection of glycerol flux could improve the ethanol yield in engineered *S. cerevisiae* by the supply increase for ethanol formation [30]. In addition, the minimization of glycerol synthesis also resulted in the yield decrease of acetic acid in *S. cerevisiae* mutant [6,7].

In the carbon flow metabolic network of *S. cerevisiae*, *GPD2*, *FPS1*, and *ADH2* were mainly responsible for the glycerol production, glycerol transport, and ethanol oxidation to acetaldehyde, respectively [8,31]. The deletion of *GPD2*, *FPS1*, and *ADH2* affected the content of ethanol during the fermentation of *S. cerevisiae*. *GPD2* and *FPS1* deletion in *S. cerevisiae* caused the decrease of glycerol content by 7.95 and 18.8%, respectively [32,33]. *ADH2* deletion in *S. cerevisiae* also resulted in the improvement of ethanol yield [34]. This study showed that the growth rate of ethanol yield in SCGFA with *GPD2*, *FPS1*, and *ADH2* deletion was higher than SCGF with double-deletion *GPD2* and *FPS1*, SCG with single-deletion *GPD2*, and SCF with single-deletion *FPS1*. Therefore, the *GPD2*, *FPS1*, and *ADH2* deletion contributed to the improvement of ethanol yield in *S. cerevisiae* based on the metabolic pathway redirection.

The maximum theoretical ethanol yield of glucose is 0.51 g (g of glucose) in *S. cerevisiae*. However, the ethanol yields only reached 90 to 93% of the maximal theoretical value in the current industrial processes [39]. In fact, during fermentation, some carbon was used to format biomass and by-products, particularly glycerol and organic acids [9]. Since even small improvements in ethanol yield would have a dramatic impact on profits in the large-scale production of ethanol, there still was a great interest to enhance ethanol yield with the formation reduction of by-products under the conditions of industrialization. In this study, SCGFA *GPD2Δ FPS1Δ ADH2Δ* exhibited the ethanol conversion rate of 0.462 g per g of glucose, which was higher than the wild-type strain (0.392 g ethanol per g of glucose). Based on the results in this study, we drew glucose metabolic pathway in *S. cerevisiae* engineering strain SCGFA *GPD2Δ FPS1Δ ADH2Δ* constructed by the CRISPR-Cas9 approach (Figure 15). The mutation sites of *GPD2*, *FPS1*, and *ADH2* were located on the glycerol synthesis pathway, glycerol transmembrane path, and ethanol consumption. These mutations resulted in the improvement of ethanol content with the reduction of by-product content. The engineering strain still maintained a high level of cell proliferation activity and glucose consumption because the gene knockout did not affect the main metabolic pathways of Embden-Meyerhof-Parnas (EMP) and tricarboxylic acid cycle (TCA cycle), as well as respiratory metabolic-related pathways. Therefore, SCGFA was suitable for industrial bioethanol production under anaerobic conditions.

## 5. Conclusions

The formation of by-products affected the ethanol yield in *S. cerevisiae* during the fermentation processing of glucose. In this study, the CRISPR-Cas9 technology was used to construct *S. cerevisiae* engineering strains by the *GPD2*, *FPS1*, and *ADH2* deletion. The highest ethanol contents of SCG, SCF, SCGF, and SCGFA increased by 5.1, 6.6, 13.3, and 17.9% compared with the wild-type *S. cerevisiae*. In addition, the glycerol contents of SCG, SCF, SCGF, and SCGFA strains decreased by 20.5, 23.1, 25.4, and 22.7% compared with the wild-type strain, respectively. SCGFA exhibited the highest ethanol conversion rate of 0.462 g per g of glucose among four *S. cerevisiae* engineering strains. According to the log_2_ (TPM) values, 166 and 404 genes respectively belonging to up-regulated and down-regulated genes were identified in the SCGFA strain. KEGG analysis indicated glycolysis, fatty acid metabolism, and carbon metabolism could affect the ethanol formation in the SCGFA strain based on the up-regulated gene enrichment. Thus, *S. cerevisiae* engineering SCGFA with the *GPD2*, *FPS1*, and *ADH2* deletion could dramatically improve the ethanol yield due to the inhibition of glycerol synthesis and the prevention of ethanol consumption.

## Figures and Tables

**Figure 1 jof-08-00703-f001:**
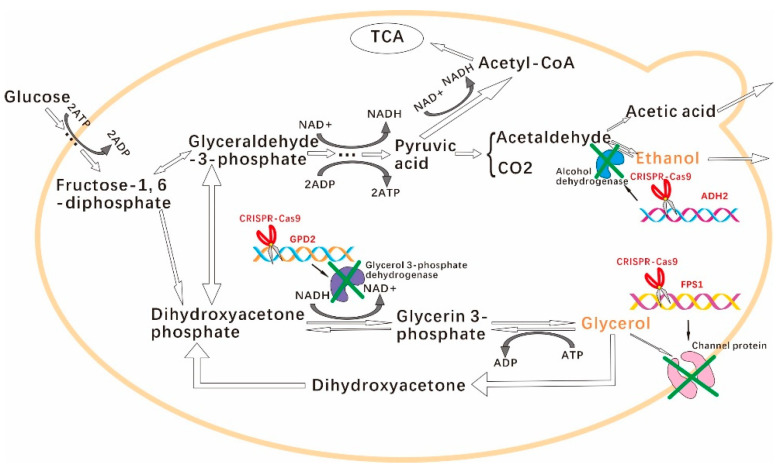
The metabolic strategies of *S. cerevisiae* in this study. GPD2 and FPS1 knockouts resulted in the decrease of glycerol accumulation. ADH2 knockout prevented the reuse of ethanol by engineered *S. cerevisiae* due to the lack of catalytic path from ethanol to acetaldehyde.

**Figure 2 jof-08-00703-f002:**
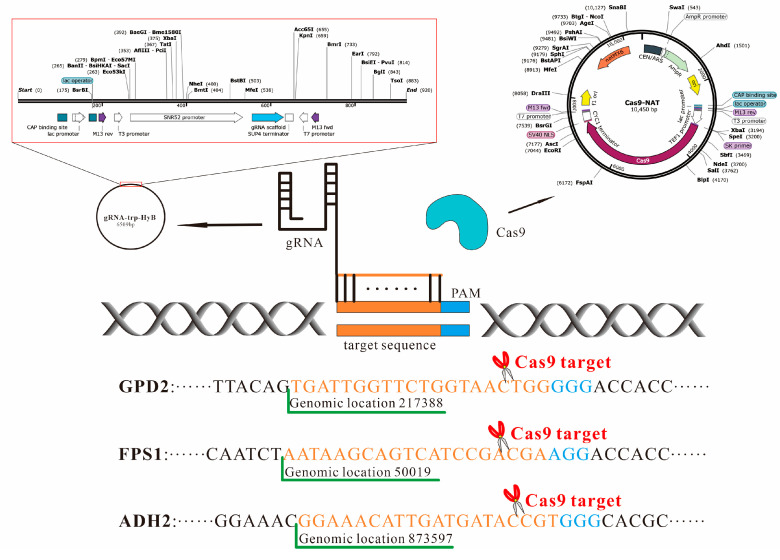
Gene knockout principle using the CRISPR-Cas9 approach. DNase expressed by Cas9-NTC cut off the genome DNA of *S. cerevisiae* under the guide of 20-bp gRNA expressed by vector gRNA-trp-HyB. The size of 20-bp gRNA was designed according to GPD2, FPS1, and ADH2 sequences. The recognition sequences of GPD2, FPS1, and ADH2 were marked in orange font.

**Figure 3 jof-08-00703-f003:**
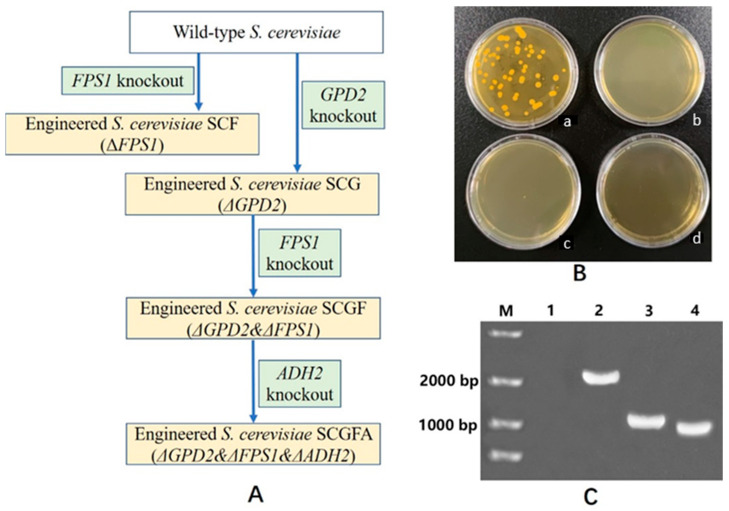
*S. cerevisiae* transformation pathways, screening, and identification. All solid YPD agar plates contained 80 μg/mL nourseothricin and 300 μg/mL hygromycin B. (**A**) Construction pathways of four *S. cerevisiae* mutants; (**B**) Screening of SCGFA *GPD2Δ*
*FPS1Δ*
*ADH2Δ* transformant. (**a**) transformant screening; (**b**) the control without addition of insert DNA; (**c**) the control without addition of gRNA-ADH2 vector; (**d**) the control using the direct culture of SCGF *GPD2Δ*
*FPS1Δ* without transformation. (**C**) PCR amplification for SCGFA *GPD2Δ*
*FPS1Δ*
*ADH2Δ* identification. M represented marker; lanes 2, 3, and 4 indicated DNA amplification bands from TV-AFB1D (2091 bp), DPE (910 bp), and OM-PLA1 (879 bp), respectively.

**Figure 4 jof-08-00703-f004:**
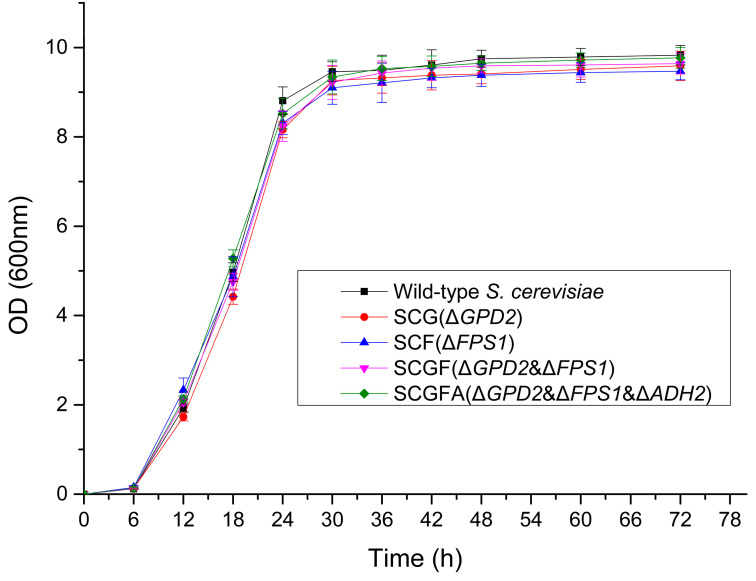
Effect of gene deletion on the growth curve of *S. cerevisiae*. SCG (*ΔGPD2*), SCF (*ΔFPS1*), SCGF (*ΔGPD2&*
*ΔFPS1*) and SCGFA (*ΔGPD2&Δ*
*FPS1&Δ*
*ADH2*) represented *S. cerevisiae* mutants with *GPD2Δ*, *FPS1Δ*, *GPD2Δ**FPS1Δ*, *GPD2Δ*
*FPS1Δ*
*ADH2Δ*, respectively.

**Figure 5 jof-08-00703-f005:**
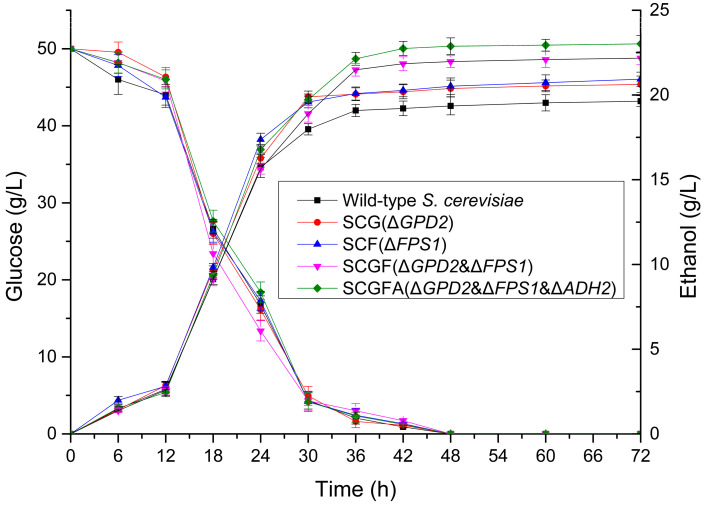
Glucose consumption and ethanol production of engineered *S. cerevisiae*.

**Figure 6 jof-08-00703-f006:**
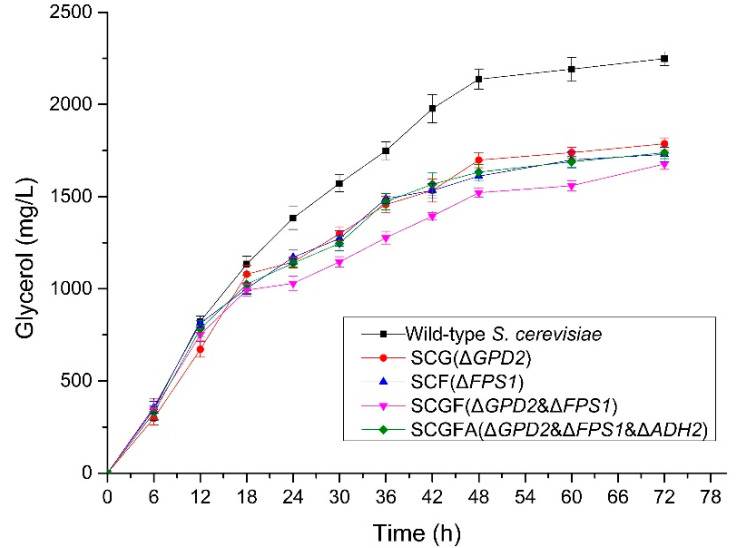
Effect of gene deletion on glycerol contents of *S. cerevisiae*.

**Figure 7 jof-08-00703-f007:**
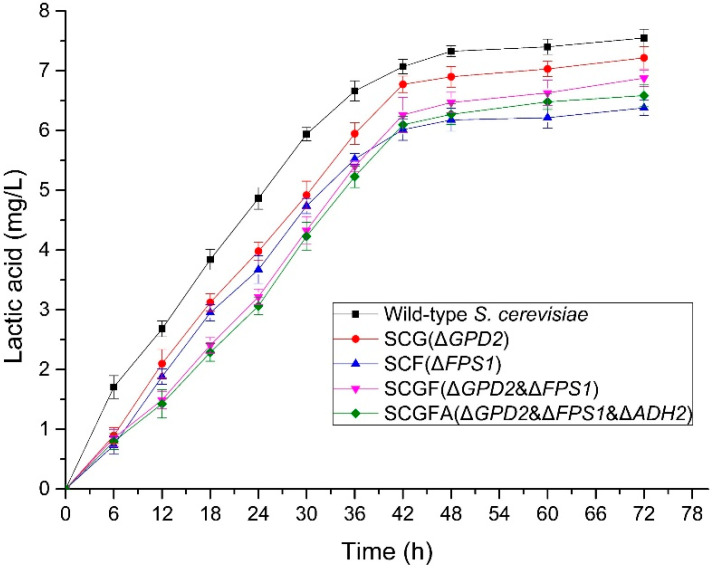
Gene deletion affecting the contents of lactic acid of *S. cerevisiae*.

**Figure 8 jof-08-00703-f008:**
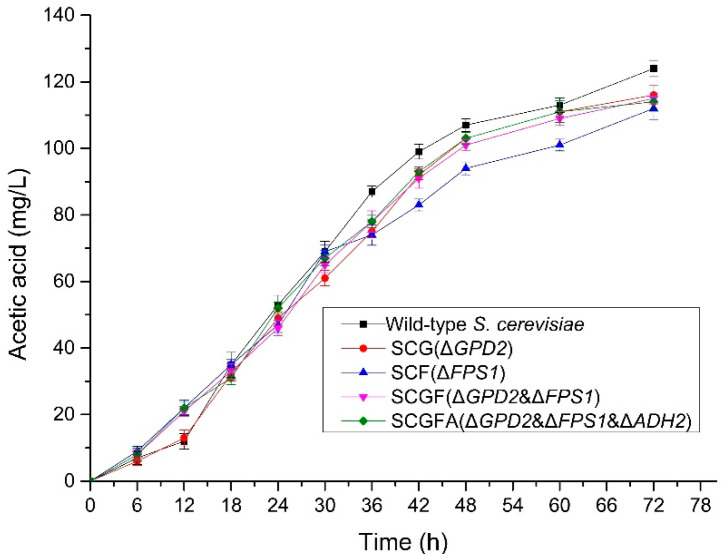
Acetic acid contents of the wild-type and *S. cerevisiae* transformants.

**Figure 9 jof-08-00703-f009:**
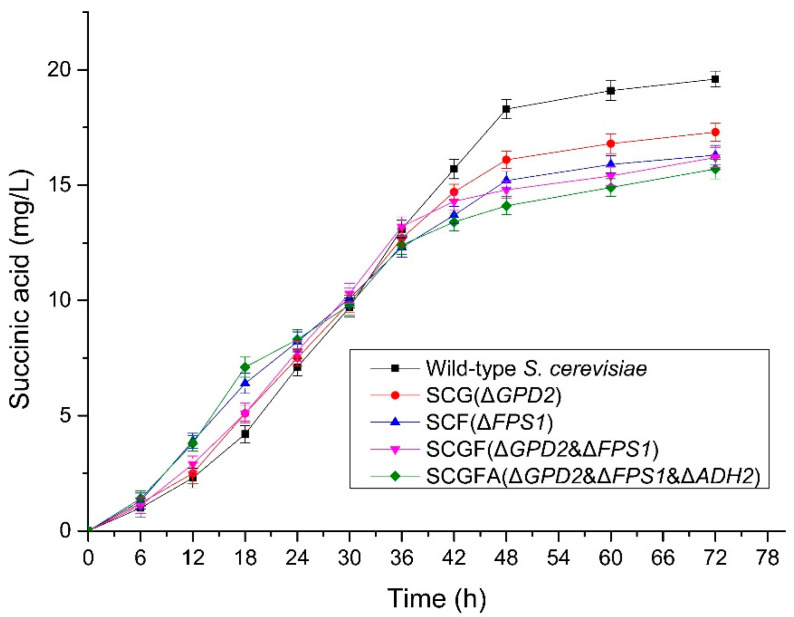
Succinic acid concentrations of the wild-type and engineered *S. cerevisiae* strains.

**Figure 10 jof-08-00703-f010:**
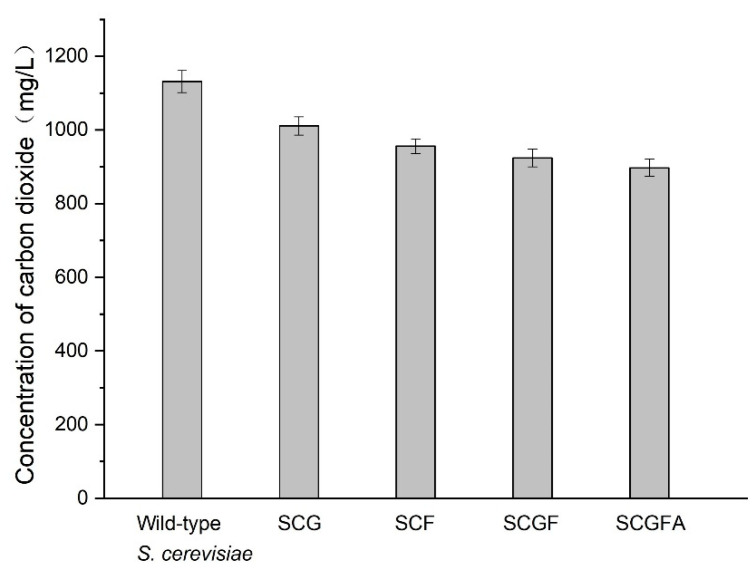
Comparison of CO_2_ concentrations between the wild-type and engineered *S. cerevisiae* strains.

**Figure 11 jof-08-00703-f011:**
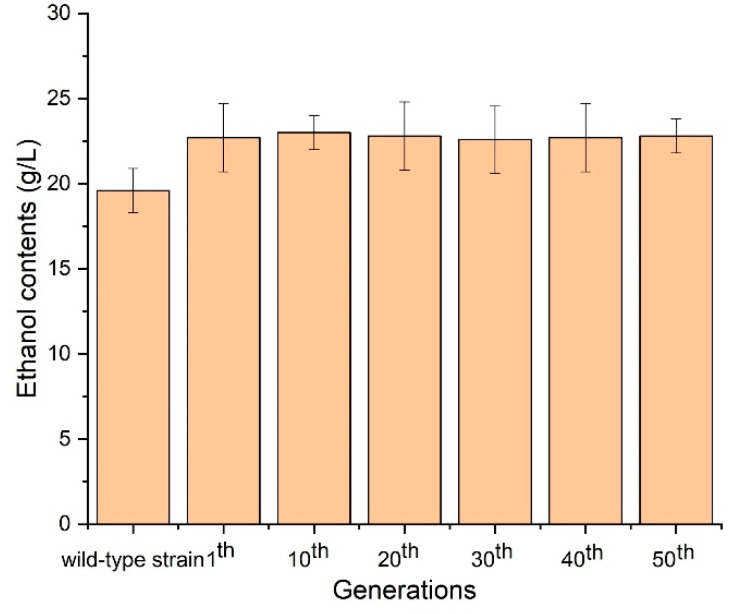
Ethanol contents of SCGFA engineering strains from multiple generations.

**Figure 12 jof-08-00703-f012:**
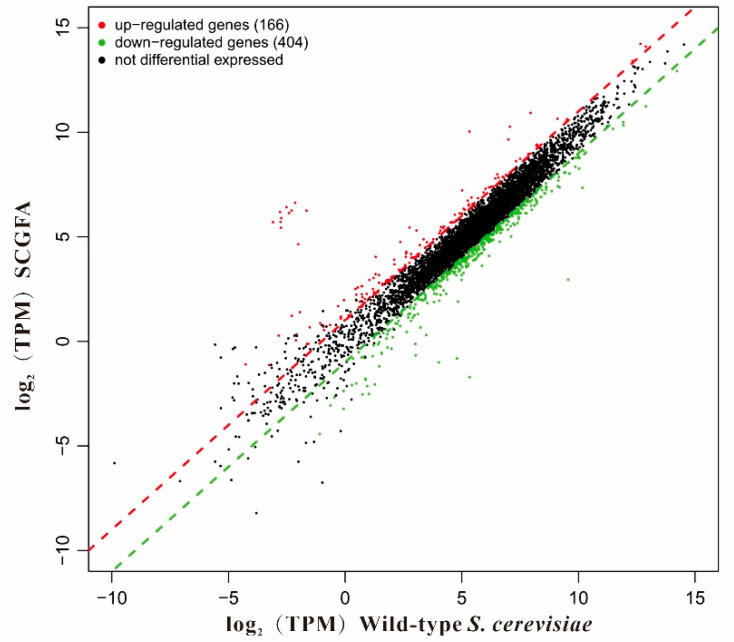
Log_2_ (TPM) values between the SCGFA and wild-type strain. The red, green, and black points represented up-regulated, down-regulated, and non-differential genes, respectively.

**Figure 13 jof-08-00703-f013:**
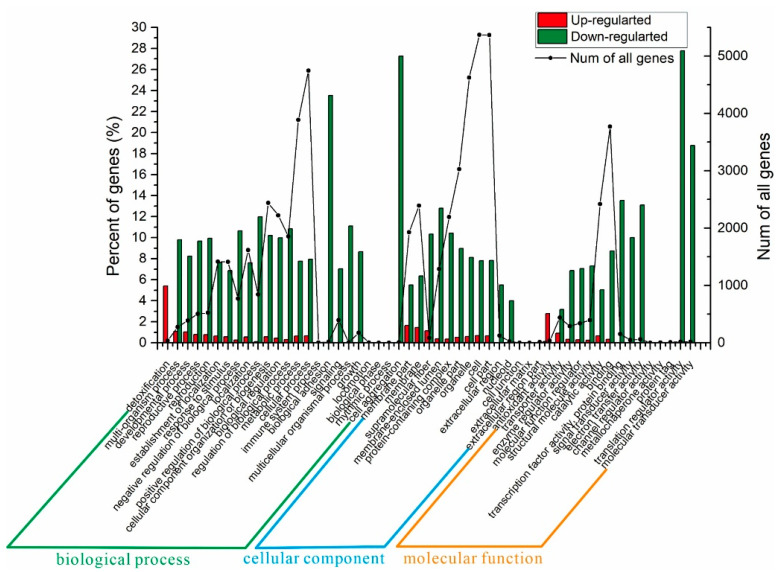
GO analysis of comparison between SCGFA and wild-type *S. cerevisiae*.

**Figure 14 jof-08-00703-f014:**
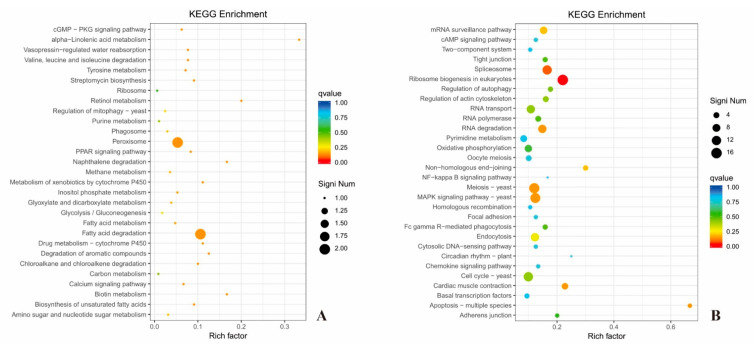
Enrichment analysis of up-regulated genes (**A**) and down-regulated genes (**B**) based on KEGG.

**Figure 15 jof-08-00703-f015:**
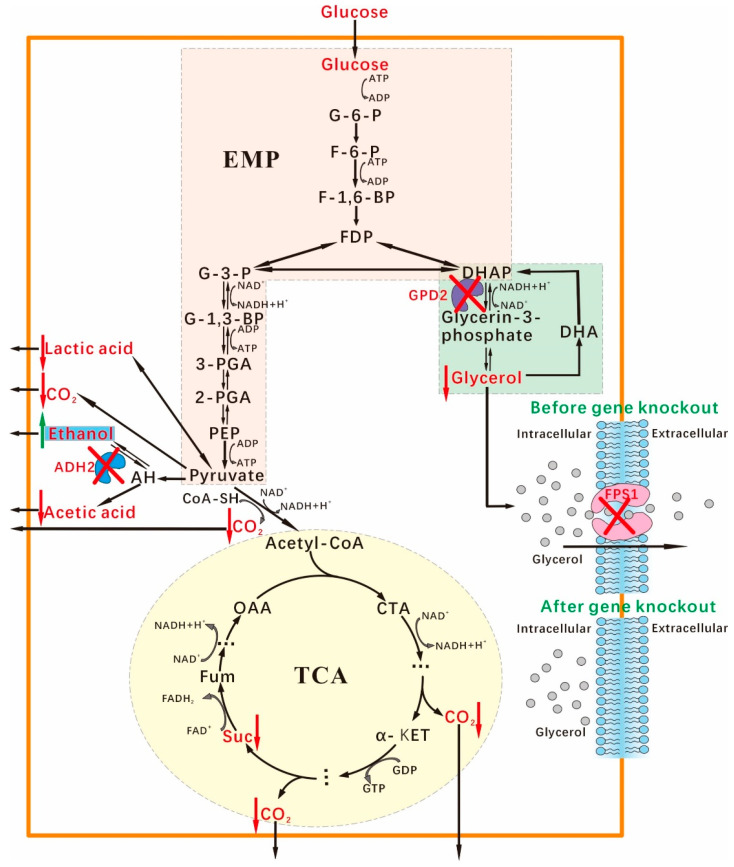
Change of glucose metabolic pathway in SCGFA mutant. SCGFA *GPD2Δ*
*FPS1Δ*
*ADH2Δ* was constructed by the *GPD2*, *FPS1*, and *ADH2* deletion using the CRISPR-Cas9 approach. The yield of ethanol was increased in SCGFA with the content decrease of the by-products of glycerol, lactic acid, acetic acid, succinic acid, and CO_2_.

**Table 1 jof-08-00703-t001:** Primers for gRNA vector construction and insertion DNA identification.

**Primers**	**Sequence**	**Description**
GPD2-gRNA-F1	**TGATTGGTTCTGGTAACTGGGGG**GTTTTAGAGCTAGAAATAGCAAG	GPD2-gRNA vector construction
GPD2-gRNA-R1	**CCCCCAGTTACCAGAACCAATCA**GATCATTTATCTTTCACTGCGGA	
Fps1-gRNA-F1	**AATAAGCAGTCATCCGACGAAGG**GTTTTAGAGCTAGAAATAGCAAG	FPS1-gRNA vector construction
Fps1-gRNA-R1	**CCTTCGTCGGATGACTGCTTATT**GATCATTTATCTTTCACTGCGGA	
ADH2-gRNA-F1	**GGAAACATTGATGATACCGTGGG**GTTTTAGAGCTAGAAATAGCAAG	ADH2-gRNA vector construction
ADH2-gRNA-R1	**CCCACGGTATCATCAATGTTTCC**GATCATTTATCTTTCACTGCGGA	
Us-TV-AFB1D	5′-ATGGCTCGCGCGAAGTACTC-3′	2091 bp
Ds-TV-AFB1D	5′-TTAAAGCTTCCGCTCTATGAA-3′	
Us-OM-PLA1	5′-TATGCGCATTTTGTCAGGGA-3′	879 bp
Ds-OM-PLA1	5′-GATTACATAATATCGTTCAGC-3′	
Us-DPE	5′-CAGAAAAGCGAAAGAGACACC-3′	910 bp
Ds-DPE	5′-TGAGGATATTATCGCAAATC-3′	

Primers of GPD2-gRNA-F1/GPD2-gRNA-R1, Fps1-gRNA-F1/Fps1-gRNA-R1, and ADH2-gRNA-F1/ADH2-gRNA-R1 were used to construct GPD2-gRNA, FPS1-gRNA, and ADH2-gRNA vectors, respectively. The underlined and bold DNA sequences were designed to amplify the target for Cas9-RNA-guided endonucleases (20 bp-NGG). The other primers of Us-TV-AFB1D/Ds-TV-AFB1D, Us-OM-PLA1/Ds-OM-PLA1, and Us-DPE/Ds-DPE were used to identify the insertion DNA with the sizes of 2091, 879, and 910 bp, respectively.

**Table 2 jof-08-00703-t002:** Contents of metabolites of different *S. cerevisiae* strains.

Metabolites	Strains (C·mol/L)				
	WT	SCG	SCF	SCGF	SCGFA
Ethanol	0.84204	0.89691	0.91013	0.96439	1.00517
Glycerin	0.07333	0.05827	0.05638	0.05468	0.05667
Lactic acid	0.00025	0.00024	0.00021	0.00023	0.00022
Acetic acid	0.00413	0.00387	0.00373	0.00383	0.00380
Succinic acid	0.00066	0.00059	0.00055	0.00055	0.00053
CO_2_	0.02570	0.02541	0.02527	0.02507	0.02489
Total	0.94611	0.98529	0.99627	1.04875	1.09128

WT represented the wild-type *S. cerevisiae*. SCG, SCF, SCGF, and SCGFA indicated four engineered *S. cerevisiae* mutants.

**Table 3 jof-08-00703-t003:** The yields of ethanol and glycerol in *S. cerevisiae* engineering strains.

Mutated Gene	Ethanol Yield	Glycerol Yield	References
*GPD2Δ*	↑5.1%	↓20.5%	this study
*FPS1Δ*	↑6.6%	↓23.1%	this study
*GPD2Δ FPS1Δ*	↑13.3%	↓25.4%	this study
*GPD2Δ FPS1Δ ADH2Δ*	↑17.9%	↓22.7%	this study
*GPD2Δ*	↑7.41%	↓7.95%	[32]
*FPS1Δ*	↑10%	↓18.8%	[33]
*FPS1Δ GAPNΔ*	↑9.18%	↓21.47%	[35]
*GPD1 ▼*	↑9.7%	↓19%	[36]
*FPS1Δ*	↑10%	↓24%	[37]
*FPS1Δ GLT1* *○*	↑14%	↓30%	[37]
*FPS1Δ*	↑3%	---	[38]

Symbol *Δ*, *○*, and *▼* represented gene deletion, overexpression, and inhibition, respectively.

## Data Availability

The datasets are available from the corresponding author on reasonable request.

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
