# Peer review of "CRISPR-Cas9 Approach Constructed Engineered Saccharomyces cerevisiae with the Deletion of GPD2, FPS1, and ADH2 to Enhance the Production of Ethanol"

_jof, 2022, doi:10.3390/jof8070703_

Round 1
Reviewer 1 Report
The authors present a manuscript on the engineering of a S. cerevisiae strain to modify ethanol production. They target three enzyme in the glycerol and ethanol pathways.
The authors use the CRISPR technology to introduce up to three mutations / deletion in the strain.
Further experiments include metabolite quantification and transcriptome analysis.
The data presented is clear and using these three knock outs is a novel way to target ethanol production in yeast. While the target genes are well known to have an impact on ethanol yield and on production of by-products, the combination as done in this work is not described before.
However, the authors fail to mention at least one relevant published work, that is the paper by Liu et al. 2019 (Biochemical Engineering Journal Volume 145, 120-126; https://doi.org/10.1016/j.bej.2019.02.017) which describes using CRISPR technology for multiple gene disruptions to engineer ethanol production in S. cerevisiae. This paper describing a very similar approach needs to be discussed in the present manuscript and novelty of the present data needs to be shown.
Some questions:
Line 362: it is an interesting observation that the down regulation of a number of genes points to impact on growth. However, this is not further analysed and discussed. A more thorough look at growth rates (not just final OD), lag time and other growth parameters would be advisable and should be added to the manuscript. Figure 5 gives some evidence of early log phase differences, but such data are not detailed.
The authors analyse organic acids such as succinic acid and acetic acid and they measure CO2. They do not present data on carbon balances nor do they discuss such. It would be good to see if carbon flow can be described as balanced or what metabolites are still missing in the analyses.
Author Response
Comment 1: The authors present a manuscript on the engineering of a S. cerevisiae strain to modify ethanol production. They target three enzyme in the glycerol and ethanol pathways. The authors use the CRISPR technology to introduce up to three mutations / deletion in the strain. Further experiments include metabolite quantification and transcriptome analysis. The data presented is clear and using these three knock outs is a novel way to target ethanol production in yeast. While the target genes are well known to have an impact on ethanol yield and on production of by-products, the combination as done in this work is not described before.
However, the authors fail to mention at least one relevant published work, that is the paper by Liu et al. 2019 (Biochemical Engineering Journal Volume 145, 120-126; https://doi.org/10.1016/j.bej.2019.02.017) which describes using CRISPR technology for multiple gene disruptions to engineer ethanol production in S. cerevisiae. This paper describing a very similar approach needs to be discussed in the present manuscript and novelty of the present data needs to be shown.
Response 1: Thanks for your professional comments, we have added the content as following: “Recently, although multiplex genome engineering has been developed to disrupt the target genes in S. cerevisiae [16-19], single genetic locus deletion is still an effective way to knock out the target gene [20].” in the Introduction section.
In addition, the novelty of this study was added in the “S. cerevisiae GPD2, FPS1, and ADH2 were involved in different metabolic pathways. The co-deletion of these three gene resulted in the ethanol content in SCGFA increased by 0.18% with the simultaneous decrease of glycerol, lactic acid, acetic acid, and succinic acid contents by 22.7, 12.7, 8.1, 19.9, and 20.7% compared with the wild-type strain, respectively.” The relevant contents existed in the Abstract section.
Comment 2: Line 362: it is an interesting observation that the down regulation of a number of genes points to impact on growth. However, this is not further analysed and discussed. A more thorough look at growth rates (not just final OD), lag time and other growth parameters would be advisable and should be added to the manuscript. Figure 5 gives some evidence of early log phase differences, but such data are not detailed.
Response 2: To detail to describe the process of cell growth, we have added the following sentences in 3.2 section. “The growth rates in the logarithmic phase of yeast proliferation were 0.4825, 0.4463, 0.4503, 0.4510, and 0.4720, respectively. The OD 600nm value and logarithmic phase growth rates of four genetically engineered S. cerevisiae were slightly lower than those of the wild-type strain after fermentation for 72 h.”
Comment 3: The authors analyse organic acids such as succinic acid and acetic acid and they measure CO2. They do not present data on carbon balances nor do they discuss such. It would be good to see if carbon flow can be described as balanced or what metabolites are still missing in the analyses.
Response 3: Thanks for your professional comments, we have added the content as following:
“3.11. Carbon balance analysis
After fermentation for 72 h, the contents of metabolites in the broth tended to be stable according to the data in Figure 4. The carbon balances of each strain were analyzed based on the contents of ethanol and the main by-products (Table 2). In this study, the concentration of 50 g/L glucose was converted into 1.66667 C·mol/L based on the molar mass of carbon. The results showed the carbon content of ethanol in SCGFA was higher than those in other strains. In addition, the total carbon content in SCGFA was 1.09128 C·mol/L, which was 1.15-fold in comparison with the wild-type strain. Thus, SCGFA exhibited a high ethanol conversion capacity.
Table 2. Contents of metabolites of different S. cerevisiae strains
|
Metabolites |
Strains (C•mol/L) |
||||
|
WT |
SCG |
SCF |
SCGF |
SCGFA |
|
|
Ethanol |
0.84204 |
0.89691 |
0.91013 |
0.96439 |
1.00517 |
|
Glycerin |
0.07333 |
0.05827 |
0.05638 |
0.05468 |
0.05667 |
|
Lactic acid |
0.00025 |
0.00024 |
0.00021 |
0.00023 |
0.00022 |
|
Acetic acid |
0.00413 |
0.00387 |
0.00373 |
0.00383 |
0.00380 |
|
Succinic acid |
0.00066 |
0.00059 |
0.00055 |
0.00055 |
0.00053 |
|
CO2 |
0.02570 |
0.02541 |
0.02527 |
0.02507 |
0.02489 |
|
Total |
0.94611 |
0.98529 |
0.99627 |
1.04875 |
1.09128 |
WT represented the wild-type S. cerevisiae. SCG, SCF, SCGF, and SCGFA indicated four engineered S. cerevisiae mutants.
Reviewer 2 Report
The MS is very relevant and interesting, and is of medium-high originality. It adds to the subject area greater precision and scientific depth compared with other published material.
All, all graphical representations MUST be improved. In this way I believe they are not publishable.
Author Response
Comments: The MS is very relevant and interesting, and is of medium-high originality. It adds to the subject area greater precision and scientific depth compared with other published material. All, all graphical representations MUST be improved. In this way I believe they are not publishable.
Response: Thanks for this suggestion. We have revised the graphical representations according to the requirements of the journal.

Reviewer 3 Report
Bioethanol is known as the most widely used biofuel. Bioethanol offers several advantages over gasoline such as higher-octane number, broader flammability limits, higher flame speeds and increased heats of vaporization. In contrast to petroleum fuel, bioethanol is less toxic, readily biodegradable and produces lesser air-borne pollutants. Microorganisms such as yeasts play an essential role in bioethanol production by fermenting a wide range of sugars to ethanol. The production of bioethanol is influenced by many factors, among which an important role belongs to the accumulation of by-products (glycerol and organic acids). An increase in the yield of biofuels can be achieved by obtaining new strains of producers with fine-tuning of biosynthesis pathways.
The presented work is devoted to obtaining yeast strains with deletions in the GPD2, FPS1 genes that control the synthesis, uptake, and efflux of glycerol, as well as in ADH2 gene responsible for the conversion of ethanol to acetaldehyde. To obtain deletions, the authors used the CRISPR-Cas9 technology.
The authors showed that the deletion of these three genes is accompanied by a higher ethanol yield, leads to a change in the transcription level of 570 genes, but does not significantly affect cell growth. Based on the results obtained, the authors conclude about the possibility of the use of the constructed strain for industrial bioethanol production.
Comments and Suggestions for Authors
1. Line 29: “…produced 23.1 g/L with 50 g/L of glucose…”.
The word ethanol is missing after 23.1 g/L.
2. Line 54: “Glycerol 3-phosphate dehydrogenase 2 (GPD2)…”
Is it talking about a protein, not a gene? To designate yeast genes, uppercase letters and italics are used, but only the first uppercase letter is used in the name of proteins, the remaining letters are lowercase, the font is normal, not italics.
3. The same remark on line 68.
4. Line 306: “…using 50 g/L as fermentation substrate…”
The word glucose is missing after 50 g/L
5. Why exactly deletion in the FPS1 gene responsible for glycerol uptake and efflux leads to the strongest decrease in lactic acid concentration?

Author Response
Reviewer 3
Comment 1: Bioethanol is known as the most widely used biofuel. Bioethanol offers several advantages over gasoline such as higher-octane number, broader flammability limits, higher flame speeds and increased heats of vaporization. In contrast to petroleum fuel, bioethanol is less toxic, readily biodegradable and produces lesser air-borne pollutants. Microorganisms such as yeasts play an essential role in bioethanol production by fermenting a wide range of sugars to ethanol. The production of bioethanol is influenced by many factors, among which an important role belongs to the accumulation of by-products (glycerol and organic acids). An increase in the yield of biofuels can be achieved by obtaining new strains of producers with fine-tuning of biosynthesis pathways.The presented work is devoted to obtaining yeast strains with deletions in the GPD2, FPS1 genes that control the synthesis, uptake, and efflux of glycerol, as well as in ADH2 gene responsible for the conversion of ethanol to acetaldehyde. To obtain deletions, the authors used the CRISPR-Cas9 technology. The authors showed that the deletion of these three genes is accompanied by a higher ethanol yield, leads to a change in the transcription level of 570 genes, but does not significantly affect cell growth. Based on the results obtained, the authors conclude about the possibility of the use of the constructed strain for industrial bioethanol production.
Response 1: Thanks for these positive and constructive comments.
Comments 2. Line 29: “…produced 23.1 g/L with 50 g/L of glucose…”.The word ethanol is missing after 23.1 g/L.
Response 2: Thanks for your professional comments, we have added the word “ethanol” in the revised manuscript.
Comment 3. Line 54: “Glycerol 3-phosphate dehydrogenase 2 (GPD2)…”
Is it talking about a protein, not a gene? To designate yeast genes, uppercase letters and italics are used, but only the first uppercase letter is used in the name of proteins, the remaining letters are lowercase, the font is normal, not italics.
Response 3: Thanks. we have revised “glycerol 3-phosphate dehydrogenase 2 (Gpd2)…”.in the manuscript.
Comment 4: The same remark on line 68.
Response 4: Thanks for your professional comments, we have revised the manuscript to “…alcohol dehydrogenase 2 (Adh2)” based on your comments.
Comment 5: Line 306: “…using 50 g/L as fermentation substrate…”The word glucose is missing after 50 g/L
Response 5: Thanks. we have revised the manuscript to “…using 50 g/L glucose as fermentation substrate…”
Comment 6 Why exactly deletion in the FPS1 gene responsible for glycerol uptake and efflux leads to the strongest decrease in lactic acid concentration?
Response 6: Thank you very much for your professional comment. The previous reports showed that fps1 Delta mutant caused the content decrease of acetic acid and pyruvic acid. (Zhang, A; Kong, Q. Effect of FPS1 deletion on the fermentation properties of Saccharomyces cerevisiae. APPLIED MICROBIOLOGY, 2007,44 (2) :212-217.). In addition, pyruvic acid is a substrate for the conversion of lactic acid. Thus, the reason of lactic acid decrease in FPS1 Delta mutant could be relevant to the reduction of pyruvic acid content. Thus, to avoid misunderstanding, we have revised the manuscript " Therefore, the deletion of S. cerevisiae GPD2, FPS1 and ADH2 resulted in the decrease of lactate content."

Round 2
Reviewer 2 Report
The manuscript describes very current and interesting research